# The Analytical Possibilities of FT-IR Spectroscopy Powered by Vibrating Molecules

**DOI:** 10.3390/ijms24021013

**Published:** 2023-01-05

**Authors:** Piotr Koczoń, Jakub T. Hołaj-Krzak, Bharani K. Palani, Tymoteusz Bolewski, Jarosław Dąbrowski, Bartłomiej J. Bartyzel, Eliza Gruczyńska-Sękowska

**Affiliations:** 1Department of Chemistry, Institute of Food Sciences, Warsaw University of Life Sciences, 02-776 Warsaw, Poland; 2Institute of Technology and Life Sciences—National Research Institute, 3 Hrabska Ave., Falenty, 05-090 Raszyn, Poland; 3Department of Morphological Sciences, Institute of Veterinary Medicine, Warsaw University of Life Sciences, 02-776 Warsaw, Poland

**Keywords:** FT-IR, ATR, chemometrics, hydrogen bonds, catalysis, quality control, medicine, greenhouse gases, food industry, forensics, museology

## Abstract

This paper discusses the state of advancement in the development of spectroscopic methods based on the use of mid (proper) infrared radiation in the context of applications in various fields of science and technology. The authors drew attention to the most important solutions specific to both spectroscopy itself (ATR technique) and chemometric data processing tools (PCA and PLS models). The objective of the current paper is to collect and consistently present information on various aspects of FT-IR spectroscopy, which is not only a well-known and well-established method but is also continuously developing. The innovative aspect of the current review is to show FT-IR’s great versatility that allows its applications to solve and explain issues from both the scientific domain (e.g., hydrogen bonds) and practical ones (e.g., technological processes, medicine, environmental protection, and food analysis). Particular attention was paid to the issue of hydrogen bonds as key non-covalent interactions, conditioning the existence of living matter and determining the number of physicochemical properties of various materials. Since the role of FT-IR spectroscopy in the field of hydrogen bond research has great significance, a historical outline of the most important qualitative and quantitative hydrogen bond theories is provided. In addition, research on selected unconventional spectral effects resulting from the substitution of protons with deuterons in hydrogen bridges is presented. The state-of-the-art and originality of the current review are that it presents a combination of uses of FT-IR spectroscopy to explain the way molecules vibrate and the effects of those vibrations on macroscopic properties, hence practical applications of given substances.

## 1. Introduction

Infrared spectroscopy is not only a current definitive tool in every laboratory dealing with basic research (i.e., related to molecular structure and aimed at understanding or predicting their properties), but FT-IR spectrometers are also used on a daily basis by researchers of any products, including food, industrial goods, and even works of art. The key to understanding the popularity of FT-IR spectroscopy is the energy dependence of atoms oscillating relative to equilibrium positions and the nature of the incident radiation beam. The condition for the transition of a molecule to an excited vibrational state is that it has at least a temporary dipole moment. Only then can the condition of radiation absorption be fulfilled.

A special place is occupied by those substances whose molecules can form hydrogen bonds, commonly found in nature. They determine the macroscopic and physicochemical properties of both inorganic substances with simple molecules (H_2_O, H_2_F_2_, and others), as well as complex organic substances (DNA, proteins, and others). Their presence is attributed to the anomalously high melting (and boiling) temperature of H_2_O and the unusual course of the change in density as a function of temperature. The basis of supramolecular chemistry—an interdisciplinary field of knowledge drawing on the achievements of organic and physical chemistry, biology and materials engineering—is hydrogen bonds.

In the context of the appearance of pollutants in fresh waters, it is very important to identify them and also to learn about the transformations they undergo after passing through the human body, as well as under the influence of naturally present substances in inland waters. Such metabolites may behave differently at different pH values, in the presence of dissolved organic matter, humic substances, or at different levels of water oxygenation or hardness. For such studies, the methods of classical organic or inorganic chemistry used to be applied, while currently, various spectroscopic and spectrometric methods are used, which are becoming faster and more accurate in operation. Most importantly, the preparation of the test sample is becoming easier, and the sample is not often destroyed in such analyses. Particularly outstanding here is FT-IR spectroscopy, which, thanks to the possibility of detecting bonds with various functional groups, allows us to not only recognise classes of compounds but also structures of chemicals and their content (e.g., in cells).

FT-IR spectroscopy is widely applied in food analysis in many areas. First of all, it allows for a multi-component, proximate analysis of food, meaning that the main nutrients (carbohydrates, proteins, and fats) and even vitamins and moisture can be simultaneously and conveniently determined using FT-IR together with attenuated total reflectance (ATR) cell technology. Infrared spectroscopy may also be used to track chemical changes that occur in food during both intentional and unintentional food processing [1]. It has also proven to be a powerful tool for screening food for adulteration and authenticity. In some cases, it can also be applied to provide information about the geographical origin of certain foods.

The aim of this paper is to show the versatility of FT-IR spectroscopy by presenting its many various analytical possibilities and also including the least obvious ones in order to enhance scientific knowledge and popularise it. Consequently, all the applications mentioned above, and more, are discussed in great detail in the following sections of this paper.

## 2. Basic Research

### 2.1. Hydrogen Bonds

#### 2.1.1. Foreword

The problem of hydrogen bonds has been central to science for many decades. In 1920, the work [2] was published, which is a milestone in the history of research in this field because its authors defined and attempted to interpret the phenomena resulting from the existence of the strongest non-covalent interactions. Although the reasons for the formation of hydrogen bonds are well-established, and the basic consequences of their implementation in the form of significant biological and physicochemical phenomena have been identified and interpreted [3,4], a number of unconventional effects resulting from the stabilisation of spatial networks of molecular crystals with hydrogen bonds still remain incompletely met. Because they determine the physicochemical properties of materials, research on their genesis cannot be limited only to the analysis of macroscopic features, including calorimetric measurements. The most effective tool of empirical research that provides insight into the course of mechanisms governing the implementation of the discussed intermolecular interactions is spectroscopy in the infrared range. Among the numerous advantages that distinguish FT-IR spectroscopy from other spectral analysis tools (X-ray or nuclear magnetic resonance spectroscopy), one should be highlighted, which is the ability to capture the dynamic nature of hydrogen bonds [5,6].

Hydrogen bonds, being a fundamental condition of biological life, remain responsible for the association of many molecular systems, both simple inorganic connections and complex organic ones [7,8]. These issues belong to the spectrum of interests in interdisciplinary sciences, drawing on chemistry and physics and focusing on engineering materials [9,10].

The spectra recorded in the mid (proper) infrared range, covering the vibration frequencies (bands) ν_X–H_, allow us to learn about the electronic structure of the associated molecules. Classic spectral effects, characteristic of bands corresponding to proton stretching vibrations of hydrogen bonds X–H···Y (Figure 1), are characterised (among others) by a shift towards longer wavelengths (red shift), as well as a significant broadening and increasing in intensity [3,4,5].

As indicated above, the formation of hydrogen bonds is accompanied by many effects, the study of which allows indirect quantification of the impact of its associations on the energy balances of analysed molecular systems. These anomalies are particularly observed during the study of colligative phenomena and transport processes [5,11,12].

In the context of studying hydrogen bonds, spectroscopy is distinguished: (i) in the (mid) infrared range (MIR); (ii) Raman (R); (iii) in the visible and ultraviolet (UV-Vis) ranges; nuclear magnetic resonance (NMR); nuclear quadrupole resonance (NQR) and inelastic neutron scattering (INS). It should be recalled that the class of spectroscopic methods does not include those widely used in classical physical chemistry and X-ray structure. Although useful, non-spectroscopic methods are not optimal tools for studying the dynamic interactions of hydrogen bonds [6,10]. Among the non-spectroscopic methods of identification of hydrogen bonds, the following currently and most widely used should be distinguished: (i) roentgenography; (ii) neutronography; (iii) calorimetry.

Historically, the first of the empirical research methods was infrared spectroscopy, and its use has contributed to a deep understanding of the nature of the sources of phenomena resulting from the formation of hydrogen bonds [2,3,4,5,6].

Infrared covers a wide spectral range (λ [m] ≈ 10^−6^–10^−2^); in this range, due to the energy values of spectral transitions, we can arbitrarily distinguish near-infrared (ν [cm^−1^] ≈ 10^3^–10^4^), proper (ν [cm^−1^] ≈ 10^2^–10^3^), and far-infrared (ν [cm^−1^] ≈ 10^1^–10^2^). Near- and proper infrared is of the greatest importance in the field of molecular systems research because the generation of oscillatory spectra is the effect of the excitation of molecules with radiation of the energies of the indicated ranges. In the case of studies of hydrogen-bonded molecular systems, this area practically covers only the proper infrared domain. Since the aim of hydrogen bond research is to understand the nature of the effects resulting from the cooperative nature of their interactions, it seems justified to analyse the spectral behaviour of their condensed systems, in particular molecular crystals. For this reason, the long-wave infrared range remains useless because its energy induces rotations of molecules [3,4,5,6,10,13].

The influence of the presence of hydrogen bonds can be identified by specific modifications of the bands, which are attributes of bond vibrations involved in the formation of bridges. Considering the oscillations of the covalently bonded atoms of the X–H groups co-forming hydrogen bonds, the spectra registered in the infrared region show [3,4,5,6]:The ν_X–H_ bands with fine, strongly developed structures;Lowering the frequency (Δν) of the ν_X–H_ bands;Increases in the intensity of the ν_X–H_ bands in the extreme (*I*_max_);Significant increases in the intensity of integral (*I*_∞_) bands ν_X–H_;Significant increases in the value of the half-widths (Δν½) of the ν_X–H_ bands.

The decrease in the value of the vibration energy ν_X–H_ is, within the harmonic oscillator approximation, a consequence of the decrease in the value of the force constants of the relevant bonds, which in turn results in an increase in their average lengths. Bond elongation, however, leads to significant deviations from the idealised description of the phenomenon. The occurrence of anharmonic vibrations affects the half-width of the associated bands. The simultaneous increase in the polarity of covalent bonds causes, apart from revealing fine structures, an increase in the area limited by their envelopes [4,6,10].

When analysing the problem of generating infrared spectra of hydrogen-bonded molecules through the prism of changes in the oscillation energy of the bridges stabilising them, vibrations other than those stretching atomic and hydrogen bonds should also be considered. The oscillations characteristic of the discussed molecular systems is summarised in Table 1 [8,10].

#### 2.1.2. Interpretation of Infrared Spectra of Hydrogen Bonds

Initially, the theories only developed in qualitative terms contributed to the knowledge of the sources of the difficulties encountered in interpretation. The breakthrough in the field of hydrogen bond research dates back to the mid-1960s. At that time, quantitative models began to be developed to describe isolated hydrogen bridges and their simple assemblies. Currently, thanks to the development of computational methods, the theoretical tools being developed make it possible to interpret and reproduce the subtle structures of the ν_X–H_ bands of hydrogen bonds’ complex conglomerates [3,4].

##### Qualitative Interpretation

Historically, the first of the qualitative theories proposed by Badger and Bauer postulated the occurrence of anharmonic couplings of stretching proton vibrations and hydrogen bonds; the characteristic broadening of the ν_X–H_ bands would be a consequence of this coupling.

An extension of the Badger and Bauer model is the theory developed by Batuev, which does not reject the concept of anharmonic coupling of low- and high-energy vibrations. This theory also assumes that the bands of hydrogen bonds present in the infrared spectra are composed of combinational subbands, with frequencies reduced or increased by the frequency of vibrations stretching the bridges. Badger–Bauer’s and Batuev’s fluctuating theories are currently only of historical importance because their predictive ability excludes obtaining results convergent with empirical results [14,15].

##### Stepanov’s Theory

The fundamental assumptions of the model proposed by Stepanov derive from the theory of electron-oscillatory spectroscopy. The energy levels of hydrogen bonds can be represented as horizontal lines plotted on parabolic curves of potential energy versus bridge length, describing ν_X–H_ vibrations in the ground and excited oscillatory states. The curves corresponding to the excited states are characterized by the occurrence of anharmonic couplings of ν_X–H_ and ν_X···Y_ vibrations by the presence of deep minima, located—in comparison with the equilibrium lengths of model hydrogen bonds—further than in the case of vibrations in the basic oscillatory state. This approach is qualitatively consistent with the Franck–Condon principle, which explains the differences in the probability of transitions between the vibrational levels of electronic states of diatomic molecules [16].

##### Bratož–Hadži–Sheppard Theory

According to the theory presented by Bratož and Hadži and independently by Sheppard, the spectral phenomena accompanying the formation of hydrogen bonds should be attributed, apart from the anharmonic coupling of proton and bridge stretching vibrations, to the Fermi resonance. The weakening of the selection rules for oscillatory transitions related to the anharmony of ν_X–H_ and ν_X···Y_ vibration affects the complexity of the structures of fine bands in the infrared spectra of hydrogen-bonded systems because they are also generated by complex tones. If the frequencies of the basic tone of a given oscillator and the complex tones of other oscillators are equal or almost equal, then a resonance occurs, which is manifested by the disappearance of the bands at the expected frequency values and the simultaneous appearance of bands with frequencies lower and higher than expected. The necessary condition for the occurrence of the Fermi resonance is that the vibrations entering the resonance belong to the same irreducible representation [17,18].

##### Hadži’s Theory

The theory by Hadži himself indicates that the energy profiles of hydrogen bonds depend on the effect of proton tunnelling through the barriers separating two potential cavities (i.e., the “double minimum” theory).

It should be noted that Hadži’s theory does not explain the nature of the mechanism of generating the bands attributed to the stretching vibrations of deuterium bonds (ν_X–D_) in the spectra in the infrared range [18].

##### Fermi Resonance Theory

The consequence of adopting the harmonic oscillator approximation is the separation of the natural vibrations of the molecules. This means that they are energy independent of each other. Since the anharmonic oscillator approximation is an adequate approach, the problem of vibration coupling should be considered.

The Fermi resonance phenomenon manifests itself in the infrared spectra of molecules whose energy levels of separate oscillators are degenerate or quasi-degenerate. However, this condition does not have to be strictly met because the vibration frequencies of the oscillators entering the resonance may be similar. The fulfilment of this condition may apply to basic tones as well as overtones or combination tones. For this resonance to appear, the contact of oscillators with the same type of symmetry must be ensured. The Fermi resonance manifests itself in infrared spectra in such a way that not one band but two bands are observed, located at frequencies lower and higher than expected [17,18].

##### Quantitative Interpretation

The development of theoretical tools for researching hydrogen bonds of a quantitative nature began less than sixty years ago. This is related to the current development of computational methods, which are the background for empirical research, whose instrumentation has also evolved. The use of equipment with advanced optics, enabling the registration of spectra with high resolution and free from non-spectroscopic interference, allowed for the study of very subtle spectral effects.

Despite considerable progress in the discussed field, it seems justified to state that the causes of some spectral phenomena cannot be satisfactorily explained within the framework of the most advanced theoretical models currently in use. This puts experimental methods in a special position, thus forcing a departure from simple, routine spectral measurements of non-crystalline samples [17,18].

##### Maréchal–Witkowski Theory

The model proposed by Maréchal and Witkowski is historically the first theoretical approach to the problem of hydrogen bonds of a quantitative nature [17,18]; This theory, although originally used to interpret the properties of the centrosymmetric dimer (CH_3_COOH)_2_, can be successfully used to explain the behaviours of both the simplest triatomic and higher hydrogen bond oligomers. It should be emphasised that the applicability of the Maréchal–Witkowski model for complex systems requires the inclusion of elements deduced by the discovery of complex spectral effects into its formalism. An undoubted advantage of this theory is the possibility of taking into account a set of spectral phenomena that have been separately considered so far (Fermi resonance, proton tunnelling, and H/D isotope exchange) [17,18].

##### Bratož’s Theory

The model given by Bratož is a tool for interpreting the hydrogen bond spectra of compounds transferred to solutions based on non-polar solvents. The factor determining the spectral properties of the tested systems in the ν_X–H_ band frequency range is the anharmonic coupling of high-energy proton stretching vibrations with hydrogen bridge stretching vibrations. These oscillations are low-energy. The ν_X–H_ bands of weak hydrogen bonds can be approximated by the Gaussian bell function [17,18,19].

##### Romanowski–Sobczyk Theory

Romanowski and Sobczyk’s model is an extension of Bratož’s theory. It is based on the assumption of a stochastic distribution of vibration energy γ_X···Y_, δ_X···Y_, and ν_X···Y_ of monomeric systems. The X···Y oscillations modify the energies of the proton vibrational transitions, which in turn affects the courses of the potential energy curves. In terms of the discussed theory, it is possible to interpret the dependence of the intensity distribution of the ν_X–H_ bands of symmetric hydrogen bond systems on the energies of the corresponding oscillations. It was noticed that in the case of strong bonds, the ν_X–H_ bands are asymmetric and broadened [20].

##### Robertson–Yarwood Theory

Robertson and Yarwood’s theoretical model is the first to explain the infrared spectral properties of weak hydrogen or deuterium bonds formed in solutions based on inert solvents. The authors point to the relaxation of ν_X–H_ vibrations as the source of the observed anomalies—broadening and blurring of the ν_X–H_ bands, which would be a consequence of the coupling of proton stretching vibrations and hydrogen bonds. The vibrations of the hydrogen bridges generate changes in the dipole moment of the X–H···Y systems, which causes fluctuations in the local electric field. These changes should be the cause of the widening of the ν_X–H_ bands, characteristic of hydrogen-bonded systems [21,22].

##### Abramczyk Theory

The proposed theoretical model assumes that two factors are responsible for shaping the fine structures of the ν_X–H_ bands. This theory was developed to explain the infrared spectral properties of weak hydrogen bonds in their collectives. It is an extension of the Robertson–Yarwood theory because, within it, the interaction of hydrogen-bonded molecules with molecules of inert solvents (i.e., “thermal bath”) is the source of the observed effects. It also takes into account the influence of changes in the polarity of bonds but neglects the effects taken into account in the previously developed models: Fermi resonance and the anharmony of hydrogen bond vibrations (ν_X–H_ and ν_X···Y_) [23,24].

##### Henri-Rousseau–Blaise’s Theory

The model proposed by Henri-Rousseau and Blaise is based on the theory given by Maréchal and Witkowski (i.e., “strong coupling”). It assumes strong anharmonic couplings of fast vibrations ν_X–H_ with low-energy oscillations ν_X···Y_ of centrosymmetric dimers of cyclic hydrogen bonds. This theory treats the interaction of the electric field of the excitation beam as a matter of a disturbance, which is the basis of the “linear response” theory. The successively developed model of Henri-Rousseau–Blaise, by analogy with the properties of the electric field, is synonymously called “relaxation” [19,25,26].

#### 2.1.3. Infrared Spectral Effects of Hydrogen Bonds

The family of unconventional infrared spectral effects for molecular crystals includes H/D isotopic substitution within hydrogen bonds and areas of molecules that are not formally involved in their formation [27,28].

The interpretation of spectral properties in terms of the Maréchal–Witkowski theory of model systems in the crystalline phase forming cyclic centrosymmetric dimers (COOH)_2_ suggests that it is not a fully adequate tool. Despite the importance of the results of spectroscopic studies of polycrystals in the frequency ranges of the ν_X–H_ and ν_X–D_ bands, the origins of some specific spectral anomalies cannot be solely interpreted on the basis of purely vibrational interactions. Their presence can be explained by assuming couplings of nucleus oscillations and electronic motions, vibronic couplings.

Spectral effects, characteristic of single crystal spectra recorded with the use of polarized radiation, are related to the temperature evolution of the ν_X–H_ and ν_X–D_ bands, as well as the non-random distribution of protons and deuterons within hydrogen bonds. The observed phenomena are a consequence of dynamic cooperative interactions [29,30,31].

##### Isotopically Neat Molecular Systems

Registration in a wide range of temperatures, using polarised radiation in the infrared region of spectra of single crystals of monocarboxylic acids forming hydrogen bond chains, allowed shedding new light on the discussed research issues.

By analysing the polarised spectra of isotopically pure molecular systems of the indicated class—containing only protons or deuterons within the bridges and fragments not involved in their formation—it can be shown that the ν_O–H_ (ν_O–D_) bands are composed of two branches with characteristic envelopes. These branches are characterised by distinctly different polarisation properties. The recording of the spectra as a function of temperature reveals special spectral effects consisting of irregular evolution of the components of the branches of these bands. Within the previously used, purely vibrational and theoretical models, it is not possible to explain deviations from the selection rules for transitions to excited oscillatory states [4,6].

##### Breaking of Vibrational Selection Rules

The attributes of transitions to the non-symmetric state (A_u_) allowed by symmetry rules are the branches of the ν_X–H_ bands, distinguished by moderately significant intensity values. In the case of the spectra of molecular systems with a tendency to form hydrogen bond cycles, the presence of anomalously intense, long-wave branches of the ν_X–H_ bands is characteristic. Theoretically, they are generated by transitions to the totally-symmetric excited state (A_g_).

For the first time, the problem of violating the rules of selection of dipole oscillatory transitions in the spectra stabilised by hydrogen bonds of molecular systems was addressed, in terms of a model dimer, over a quarter of a century ago [32]. The most important sources of anomalous effects are vibronic couplings, qualitatively similar to those determining spectral properties in the ultraviolet and visible ranges of molecular systems containing coupled π orbitals. The mechanism responsible for the promotion of forbidden transitions in the infrared range is a kind of “reversal” of the vibronic Herzberg–Teller mechanism. It should be noted that the proposed model allows for the interpretation of the infrared spectral properties of molecular crystals, in the lattices of which dimers stabilised with N–H···S=C bridges can be identified [30,33,34].

##### Variability of Spectral Generation Mechanisms as a Function of Temperature

Despite the significant development of quantitative theoretical models, issues related to the mutual interaction of hydrogen bonds are an important research problem. The effects observed in the infrared spectra of molecular systems containing centrosymmetric cyclic hydrogen bond dimers have been recently elucidated. This concerns, among others, the dependence of the intensity distribution of the ν_X–H_ bands on the electronic structure of associates and temperature [31].

In the case of monocarboxylic acids and selected dicarboxylic ones (Figure 2), whose molecules contain aliphatic, unbranched carbon skeletons, the ν_O–H_ bands of the spectra recorded at high temperatures (*T* [K] ≈ 298) are characterised by the presence of long-wave branches with low intensities compared with high-energy branches. Lowering the recording temperature of the spectra to the boiling point of liquid nitrogen (*T* [K] ≈ 77) is only accompanied by a slight increase in the intensity of the long-wave branches, but they do not remain more intense than the high-energy branches. The spectral properties of acids in the molecules of which the carboxyl groups are separated from the aromatic centres by methylene groups are qualitatively similar to those of aliphatic compounds belonging to the indicated class. This applies to high- and low-temperature spectra [35].

The analysis of infrared spectra of monocarboxylic acids, whose functional groups are located in the immediate vicinity of extensive π-electron cores, including temperature effects in the frequency range of ν_O–H_ bands, clearly indicates a strong influence of the electronic constitution of the associating molecules on the spectral properties of hydrogen bonds. This influence manifests itself in the form of a different course of the temperature evolution of the component branches of the bands (which are attributes of proton stretching vibrations) compared with the acids belonging to the previous groups. It turns out that the low-energy branches of the ν_O–H_ bands of high-temperature spectra are characterised by high values of relative integral intensities. The ratios of the intensity of the long-wave branches to the short-wave branches increase with the lowering of the sample temperature close to the boiling point of liquid nitrogen.

The spectral properties of the hydrogen bond chain systems remain roughly different compared with those previously discussed. In the case of systems containing aromatic cores in the immediate vicinity of hydrogen bonds, the intensity distribution of the branches of the ν_O–H_ bands recorded in infrared spectra is qualitatively similar to that of aliphatic, unbranched carboxylic acids forming (COOH)_2_ cycles. In the case of chain systems built of molecules that do not contain easily polarisable electrons (except for the π orbitals of carbonyl and thiocarbonyl groups), the spectral properties are not qualitatively similar to those of simple, saturated carboxylic acids that form (COOH)_2_ dimers. The evolution of the branches of the ν_O–H_ bands is extremely and strongly dependent on the temperature [27,30,36].

Previous works devoted to the problem of model dimers of hydrogen bonds led to the conclusion that the (COOH)_2_ cycle should be considered the most important carrier of spectral properties in the infrared range. In light of the results of the research carried out so far, this argumentation does not seem to be fully unjustified. Its rejection is supported by the fact that there was no polemic related to the vibronic effects [5].

##### Hydrogen Bond Systems Isotopically Diluted with Deuterons

The spectral consequences of replacing protons with deuterons within hydrogen bridges were only considered in the aspect of not going beyond the purely vibrational approach, thus only taking into account the mass factor. This procedure seemed to be acceptable because (according to theories recognised at the time) it was believed that the concentration of cyclic or chain proton and deuteron bridges in molecular crystal networks was determined solely by the rules of stoichiometry, and the mutual orientation was subject to a normal distribution [4,6].

The analysis of infrared spectra of isotopically pure and diluted molecular crystals with deuterium atoms leads to the conclusion that protons and deuterons are subject to non-stochastic distribution within hydrogen bonds of crystals with variable H/D isotopic composition. This fact is confirmed by the significant invariance of the ν_X–H_ and ν_X–D_ bands. The discovery of H/D isotopic “self-organisation”, which is a consequence of the dynamic cooperativeness of hydrogen bridges, is a very significant contribution to the development of modern chemical physics [6,27,28,30,31,32,33,34,35,36,37,38,39].

##### The Isotopic H/D “Self-Organisation” Effects

The generally accepted theory, proposed in the 1960s and developed in the following decade, considered the distribution of protons and deuterons in crystal lattices to be completely random. The empirical foundation of the proposed theory was to be the infrared spectra of single-crystalline preparations of selected homologous *n*-alkan-1-ols (*n*-alcohols), the molecules of which were subjected to isotopic dilution with deuterons (C_2_H_5_OD, C_6_H_13_OD, C_9_H_19_OD, C_10_H_21_OD, C_11_H_23_OD, C_16_H_33_OD; Figure 3). Compounds belonging to the analysed class do not show a tendency to H/D isotope anomalies, which is spectrally evidenced by the presence of single, narrow signals in their spectra located in the middle of the ν_O–H_ bands [3,5].

The random distribution of protons and deuterons within the crystal lattices of the spectroscopically analysed alcohols seems to be a sensation in the light of the results of studies of complex hydrogen bond systems, within which their chains, centrosymmetric cyclic dimers (or larger groups) can be distinguished. The exchange of protons for deuterons leads to the generation of ν_X–H_ (“residual”) and ν_X–D_ bands as an equilibrium process. Since the presence of ν_X–H_ bands in the infrared spectra is a consequence of deuteron elution during crystal growth; thus, it seems advisable to analyse their temperature and polarisation properties against the background of ν_X–D_ bands. The indisputable invariance of the specified properties of the “residual” ν_X–H_ bands as a function of deuteron concentration makes us strongly reject the previous theoretical approach. It should be emphasised that it was, therefore, unwise to select homologous *n*-alcohols as representative molecular systems [6,37].

The registration of infrared spectra of isotopically diluted molecular crystals in cyclic associations of functional groups allows us to state a certain regularity characterising the distribution of protons and deuterons. Only (COOH)_2_ (“HH”) (Figure 4) or (COOD)_2_ (“DD”) (Figure 5) dimers are present in the crystal lattices of the indicated molecular systems. Mixed dimers (COOH)(COOD) (“HD”) are undetectable by infrared spectroscopy. Molecule fragments within which protons and deuterons cluster in a non-random manner are called “domains”. Based on the results of previous studies, it seems that the H/D isotopic “self-organisation” is a phenomenon characteristic of systems forming centrosymmetric hydrogen bond dimers [6,30,37].

The explanation of the phenomenon of “self-organisation” of the H/D isotope should be carried out on the basis of thermodynamic considerations because the values of free formation enthalpies of cyclic and chain dimers “HH” and “DD” differ [8]. It should also be emphasised that, in the study of the spectral properties of hydrogen-bonded systems (in particular, the discussed H/D isotopic “self-organisation”), using the tools of theoretical chemistry is impossible. This is an obvious consequence of the separation of electronic and nuclear motions (Born–Oppenheimer approximation). The theoretical mechanism describing the unconventional H/D isotope effects should assume the occurrence of couplings as a vibronic nature, including ν_X–H_ (ν_X–D_) vibrations and electronic movements within the bridge cycles [4,6,27].

The conclusions resulting from the analysis of the values of the appropriate thermodynamic functions for the considered molecular systems confirm the impossibility of spectral identification of dimers with a mixed isotopic composition (“HD”). The processes of H/D isotopic “self-organisation” are a consequence of dynamic cooperative interactions. Energy balances of the formation of only symmetric dimers of hydrogen bonds (“HH”) or deuterium (“DD”) dimers should contain specific increments, the presence of which is a consequence of the occurrence of fully symmetric vibrations within the discussed dimers, and the vibration energy values ν_X–H_ of the “HH” and ν_X–D_ of the “DD” associations differ. The additional generated stabilisation energy of isotopically pure dimers makes a significant contribution to their total formation energies. The energy of dynamic cooperative interactions may constitute 15% of the energy of interactions [27,30].

##### “Long-Range” H/D Isotopic Effects

The discovery and explanation of the nature of the influence of isotopic substitution (H/D) of hydrogen-bonded molecules of molecular crystals on their spectral properties in the infrared range are one of the most important achievements of modern chemical physics. It was noticed that the presence of protons and deuterons within the carbon skeletons modifies the fine structures of the ν_X–H_ and ν_X–D_ bands. The source of the newly discovered H/D (“long-range”) isotope effects is undoubtedly the coupling of vibrations ν_X–H_, ν_X–D_, ν_C–H_, and ν_C–D_ with electronic motions within the associations [6,27,36].

The analysis of infrared spectra of systems with a mixed isotopic composition —within hydrogen bonds and fragments not involved in their formation—shows that the presence of only protons or deuterons in associates induces anomalous spectral properties of the ν_X–H_ and ν_X–D_ bands (Figure 6). The stability of the isotopic composition of molecules (exhaustive substitution with protons or deuterons) determines the implementation of the forbidden transition promotion mechanism, which is weakened along with the progressive isotopic dilution (D or H). It has been proven that the modification of the isotopic composition results in a change in the electron density distribution of the associated molecules. On the basis of infrared spectroscopic studies of molecular crystals in which isolated dimers of hydrogen bonds or their chains are present, it can be concluded that the saturation of carbon skeletons with protons and the introduction of deuterons into the bridges results in a significant narrowing of the ν_X–D_ bands. An analogous modification of the ν_X–H_ bands is favoured by the hydrogen bonding of molecules with perdeuterated backbones. No differentiation of the spectral properties of the discussed systems is observed during the recording of low-temperature polarised spectra. The “long-range” H/D isotopic effects are most pronounced in the spectra of systems in which π orbitals are present, stretched between atoms directly adjacent to the hydrogen bonds of cyclic or quasi-chain dimers [27,32].

### 2.2. Catalysis

Works on the use of spectroscopy in the study of the course of catalytic reactions are devoted to both organic [40,41,42] and inorganic [43,44,45] processes.

In the study of the mechanisms of catalytic processes of hydroformylation reactions [40], the transmission method of measuring IR spectra and the ATR technique was used with the detection of substrates and products in a batch reactor system and a continuous operation mode under conditions corresponding to large-scale industrial synthesis. Time-dependent IR spectra were collected and processed using the BTEM (band-targeted entropy minimisation) algorithm. Simultaneous chromatographic (GC) analyses confirmed the agreement in terms of the concentration profiles of all organic compounds.

In a broader context, the study of catalytic reactions includes processes that have been used in industry for a long time, including the Ziegler–Natta reactions, dehydrogenation, and dehydration, and numerous processes based on chemisorption (Na/V_2_O_5_/TiO_2_-ZrO_2_, NO-O_2_/Tb_4_O_7_-La_2_O_3_ and others) [41,42,45].

FT-IR spectroscopy is also used to study inorganic processes, including the reduction of non-metal oxides. For the selective catalytic reduction of NO on Cu-substituted chabazite (zeolite), time-resolved spectroscopic measurements were used in the context of the analysis of the composition of the analysed gases (NO_x_ and NH_3_) within the deposit. It was found that a catalyst containing Cu-active sites, mainly bound to zeolite, selectively adsorbed NH_3_, and the temperature-dependent kinetic pathway of this compound was confirmed [43].

In the context of the rich chemistry of boron compounds, it is necessary to monitor the presence of BO_3_, the health impact of which is not indifferent. FT-IR turns out to be a sensitive method for distinguishing the structure of boron trioxide and borates at the level of the acceptable detection limit of 0.1% by weight [44].

## 3. Technological Processes

### 3.1. Product Quality Control

The use of infrared spectroscopy for the purposes of product quality control is very wide. It concerns not only utility products [46] but, most often, food products [47,48,49,50].

In the context of the problem of analysing the pigments of two-component epoxy paints, the universal role of FT-IR spectroscopy becomes apparent, which can be successfully used for the analysis of such simple inorganic compounds (Fe_2_O_3_ and Cr_2_O_3_) as well as complex organic systems (porphyrins) [46].

The authors of the work [47] drew attention to the problem of flour quality, primarily in the context of factors important from the point of view of nutritional value (protein, moisture, and fat content). Reports in the literature indicate that the content of fatty acids is often analysed in nuts [48].

Interesting are the studies on the qualitative composition of herbal mixtures used in traditional Far Eastern medicine, which results in particular from the complexity of the raw materials used in preparations. This fact translates not only into difficulties in assessing the usefulness of use but also in classifying the constituent substances as having therapeutic activity. The authors [49] show the importance of measurements of correlated spectra (2D-IR) against the background of the use of non-spectroscopic methods (chromatography). Among the advantages of spectroscopy in the mid-infrared (proper) range, the possibility of comprehensive determination of a given preparation’s composition, including undesirable components whose therapeutic value is neutral or negative, should be indicated. Another advantage compared with other methods is the speed and simplicity of implementation, without the need for tedious sample preparation. This is reflected in the use of spectroscopic techniques in the pharmaceutical industry, where extensive calibration models are used to monitor the quality of products [49,50].

### 3.2. Installation Tightness Test

Although the amount of literature testing and reporting the quality of technological lines in the context of corrosion control with the use of infrared spectroscopy is not significant, it seems that this branch of engineering is a promising niche in which the advantages of the discussed technique can be shown [51,52,53].

Particularly noteworthy is the use [51] of the innovative MEMS-FTIR method to identify the distribution of Al_2_O_3_ nanoparticle clusters exposed to UV radiation in order to provide model conditions for material aging.

In order to protect transmission installations exposed to the adverse impact of external conditions, they are often subjected to cathodic protection with the use of polymer coatings as additional protection. In work on this issue [52], the strength of a copolymer based on high-density poly(ethylene) (HDPE) was assessed. It has been proven that, as a consequence of the cessation of cathodic protection, moisture penetrating the installation does not lead to degradation of the protective material for at least 30 days.

Research in the field of protective materials engineering, in which FT-IR spectroscopy is used, also comes down to its use as an auxiliary tool to a marginal extent. In work devoted to joining car bodies, the use of FT-IR spectroscopy was only limited to the initial qualitative assessment of the composition of the binder preparation [53].

## 4. Medicine

IR spectroscopy is often used in the context of research in medicine, biotechnology, and pharmacy [54,55,56].

An interesting use of the ATR technique is provided in [54]. The authors constructed biosensors based on a layer of long-chain esters docked on a semiconductor substrate, capable of binding proteins, which in turn can bind to antibodies.

A serious application of FT-IR spectroscopy is blood tests in the context of analysing hormones [57], proteins [55,58], and glucose [59]. These are often interdisciplinary studies, drawing on methods used in computational chemistry (molecular mechanics) [58], in addition to routine methods of discriminatory analysis of results [55].

## 5. Environmental Protection

### 5.1. Research of the Greenhouse Gas Fluxes

The systematic increase in the concentration of greenhouse gases in the atmosphere associated with human economic activity results in global climate changes. Mitigating these changes is currently one of science and practice’s greatest challenges.

One of the sources of greenhouse gases is ecosystems that have been transformed to varying degrees as a result of human activity. Undertaking measures to reduce the emissions from these sources requires the recognition of the magnitude and dynamics of these gas fluxes. In order to take these actions, it is also important to know how the magnitude of these fluxes change under the influence of human activities and know the potential to reduce these gas emissions from individual ecosystems into the atmosphere. Currently, measurements of greenhouse gas fluxes in various types of ecosystems are carried out all over the world. Most often, researchers are interested in CO_2_, CH_4_, and N_2_O fluxes, which are among the most important greenhouse gases [60].

Gas analysers using FT-IR spectroscopy (apart from devices based on the ND-IR method and the photoacoustic method) are used in the study of greenhouse gas fluxes. Gas analysers using the FT-IR technique allow simultaneous measurement of concentrations of many gases, including gases present in very low concentrations. The time of measurement performed with this type of device is relatively short. These possibilities are particularly desirable in this type of research

The chamber method is the first to measure greenhouse gas fluxes using the FT-IR technique of gas concentration measurement [61]. It is one of the most widely used measurement methods in the world [60,62]. Measurement using the chamber method consists in closing a part of the ecosystem in a sealed chamber and recording the rate of changes in the concentration of the gas inside the chamber over a specified period of time [63]. Depending on the construction of the measurement system, the gas analyser may be located inside the chamber or outside the measuring chamber (closed system: chamber–analyser–chamber) [64].

An increase in the concentration of the gas in the chamber indicates a flux directed from the tested surface to the atmosphere, while a decrease indicates a flux directed in the opposite direction [65].

The chamber method enables the measurement of gas exchange between the ecosystems and atmosphere or between the soil and atmosphere. It is also used in laboratory conditions (e.g., in testing the biological activity of soil samples). This method is characterised by a small research scale, reaching a single plant or even the size of a single leaf [63,64,66]. The application of this method makes it possible to know the spatial diversity of gas fluxes within heterogeneous ecosystems (e.g., with the distinction of specific patches of vegetation) [67,68,69]. These methods allow measurements to be made in locations where other methods cannot be used [63].

The basic advantages of chamber methods, which determine their widespread use, are the relatively low cost of measurement systems and their simplicity [70]. Chamber measurement systems are characterised by high mobility and ease of installation in the field [63,69]. The results obtained using the described method can be used to supplement the series of measurement data obtained with other methods in conditions where these methods do not allow for obtaining reliable results. Another advantage of this method is the ability to determine the amount of net CO_2_ exchange of the ecosystem along with its individual components (e.g., soil respiration, total ecosystem respiration, and gross ecosystem production).

The small scale of the area and volume covered by chamber methods, which is an advantage in some studies, is a serious limitation in the case of research on ecosystems covering a large area and ecosystems with high vegetation. Therefore, the use of chamber methods is limited to ecosystems with low vegetation [71] or periodically devoid of plants [72]. Examples of such ecosystems are the ecosystems of agricultural crops: permanent grasslands and ecosystems of field crops (on arable land). In the case of forests, this method can be used to measure gas fluxes in the lowest floors of these ecosystems [69].

Chamber methods require the fulfilment of numerous, often difficult conditions despite the fact that they are based on simple methodological foundations. Their use requires special attention and care. Otherwise, there may be many sources of measurement uncertainty (e.g., leaks in the measuring system, change in pressure inside the chamber when placing the chamber on the soil frame) [73,74].

Another method of measuring greenhouse gas fluxes in which FT-IR gas analysers are used is the profile method. The basis of this method is the Monin–Obukhov similarity theory. Selected scalar quantities (e.g., CO_2_ concentration) and horizontal components of wind speed of at least two heights above the tested active surface are measured here [64]. Gas analysers are most often installed on special measuring towers.

The main advantage of the profile method is the ability to estimate the magnitude of net exchange gas fluxes on the scale of entire ecosystems [75]. One of the main advantages of the profiling method is its simplicity [64]. This method does not require the use of the fastest available measuring devices, which significantly reduces the cost of building a measuring system. Measurements with this method can be performed regardless of the height of the vegetation in the ecosystem and are carried out without disturbing the vegetation and soil.

The profile method is only applicable in homogeneous ecosystems with a sufficiently large area, and it only allows one to determine the magnitude of the net exchange of given gas between the ecosystem and atmosphere. A difficulty in applying the profile method is the need to apply appropriate corrections to the flux value in the case of surfaces characterised by significant roughness coefficient values (e.g., tree canopies) [75].

Compared with other methods, the measurement systems used in the profile method are relatively cheap to maintain [64,76,77].

The choice of the measurement method used depends on the type of ecosystem under study, the requirements of the research team, and the available resources. Each of these methods has advantages as well as limitations. The described methods can complement each other, and both methods can be used to control the accuracy of the obtained results mutually.

### 5.2. Flora and Fauna of Water Areas

Microalgae are species of mostly single-celled algae and aquatic plants that most often float in the water column. These include, among others, green algae and cyanobacteria (which, according to the last classification, belong to Protista) [78]. These organisms, due to their rapid multiplication in optimal environmental conditions, are considered to be the basis of the trophic pyramid [79]. For this reason, research on this group of plants is very often performed. In the context of the increasingly strong eutrophication of fresh waters with the simultaneous increase in water temperature, it is important to thoroughly understand the ecological requirements of algae, as well as their competitive abilities. The N:P:C ratio, which is inappropriate for a given group of algae, causes the displacement of some taxonomic groups of algae by others. Another important problem in inland waters is the presence of pharmaceuticals, endocrine-activating substances, and personal care products [80]. They have a variety of effects on aquatic organisms, often depending on the species.

For research on the aquatic environment, both FT-IR and UV-Vis spectroscopies are often used. An example of such research is [81]. The authors compared gravimetric and FT-IR methods for the determination of lipids, carbohydrates, and proteins in microalgal cells used for the production of biodiesel (*Chlorella pyrenoidosa, Nannochloropsis* sp., *Chlorella vulgaris, Chlorella vulgaris* bey (FACHB-1227), *Botryococcus braunii* (FACHB-357), *Microcystis aeruginosa* (FACHB-940)). Differences in the results between gravimetric methods and FT-IR were found, probably caused by chlorophyll dissolved in methanol during ultrasonic extraction (higher values were reported from gravimetric methods). In the case of lipids, the authors noted higher values in the FT-IR than the real ones, which was caused by the research hypothesis that the algal biomass only consists of lipids, proteins, and carbohydrates, which is obviously an incorrect assumption. According to the authors [81], the FT-IR method is preferred for analysing the composition of algae biomass (of course, with the correct assumption that not only lipids, proteins, and carbohydrates make up the biomass of these organisms). The fluorescence spectroscopy after Nile red staining mentioned in the paper indicates an interesting application of this method, namely the determination of the location of lipids in the algal cell, which can be used in cell research. Similar issues, only in the case of proteins (without lipids and carbohydrates), were dealt with by [82]. These authors used both FT-IR and near-infrared (NIR) spectroscopy. They raise the issue of the difficult use of UV spectroscopy for this type of research due to the influence of nucleic acid residues, buffers, and impurities formed during extraction (the problem is the thick cell wall protecting plant cells, which must be broken to gain access to the inside of the cell). Another problem with UV spectroscopy is that the absorbance at 280 nm is dependent on tyrosine and tryptophan, as well as phenylalanine (to a lesser extent). According to [82], this problem does not exist in FT-IR spectroscopy because it contains spectral lines for functional groups of proteins, fatty acids, and carbohydrates, which makes it possible to use them for quantitative analysis. According to [82], after trimming the spectral spectrum to the range of 800–1800 cm^−1^ in FT-IR spectroscopy, the band of amide I centred around 1650 cm^−1^ was used, with the overlap of this spectral line with bands of arginate and uronic acid polysaccharide (from the brownstone walls). A better solution was the spectral line of amide II appearing in the range of 1485–1565 cm^−1^ (from the bending vibrations of the −NH and −CN groups).

As shown by [83], it is possible to use FT-IR spectroscopy in the micro version to study the biomass composition of algae subjected to food stress (starvation). Five species of algae, namely three blue-green algae (*Microcystis aeruginosa*, *Chroococcus minutus*, and *Nostoc* sp.) and two species of diatoms (*Cyclotella meneghiniana* and *Phaeodactylum tricornutum*), were studied. For these analyses, spectral lines corresponding to proteins, lipids, carbohydrates, and additionally (for diatoms) silicates were used. The validity of such experiments has been demonstrated, indicating that the ability to cope with food stress is specific to each of the tested species, and experiments should be carried out on single cells. Absorption in the range of 400–1800 cm^−1^ was also used here. Reference [83] also indicated that wavenumber ranges spectra of particular classes of compounds should be sought (Table 2).

The results presented by [83] are discussed in an interesting way and confronted with the results of FT-IR, although they are generally known to hydrobiologists as certain dogmas, such as the C:N, P:N ratio [80], which favour the appropriate groups of phytoplankton. These studies can be regarded as revolutionising plant physiology in hydrobiology and make it possible to predict (with varying sensitivity) the content of phosphorus in water based on appropriate spectra corresponding to this element. In addition, it would probably be possible to determine the estimation of the level of nutrients based on coefficients describing biomass based on statistical analyses.

Another application of FT-IR is reported by [84,85]. These authors suggest using this method to study the effect of two antibiotics: chloramphenicol and roxithromycin, on four species of algae (green algae: *Pseudokirchneriella subcapitata, Scenedesmus quadricauda, Scenedesmus obliquus*, and *Scenedesmus acuminatus* [84]), and study the effect of carbamazepine on *S. obliquus* and *Chlorella pyrenoidosa* [85]. Analyses carried out by the authors of [84] indicate that both antibiotics have a negative effect on the growth of the tested algae, and the effects differed depending on the species, with *S. acuminatus* turning out to be the least sensitive. Comparing both antibiotics showed that chloramphenicol turned out to be less toxic than roxithromycin, though both of these antibiotics act on algae by inhibiting fatty-acid synthesis and promoting protein and DNA aggregation. In turn, carbamazepine [85] affected superoxide dismutase (SOD) and catalase (CAT), and the production of chlorophyll A. Carbamazepine had a stronger effect on the production of chlorophyll A and catalase for *S. obliquus*, and in the case of *C. pyrenoidosa*, carbamazepine had a greater effect on superoxide dismutase. An important conclusion from this study is that carbamazepine affects tested organisms more strongly during long-term action than during short-term action of higher doses.

FT-IR is a completely different method than UV-Vis and fluorescence spectroscopy (which in turn has more applications, e.g., in determining the origin of chemical compounds in the sediment). Certainly, both methods are worth looking at and applying for further research in the field of hydrobiology, such as the further movement of various compounds and elements (e.g., antibiotics and heavy metals) in food webs. If we can detect the origin of these substances and study how they affect unicellular algae, it should be possible to study what effect zooplankton would have on unicellular algae, which were previously cultured in water with various antibiotics. The next step would be to explore more trophic levels. It should be remembered that harmful substances accumulate in food webs due to the ever-larger areas needed by subsequent consumers.

In the aquatic environment, organic matter exists in a number of states, generally referred to as DOM (dissolved organic matter) [86]. This matter is the basis of many food chains in both stagnant and flowing waters, as well as in the seas, as it is food for many organisms, such as microorganisms [87] or macrozoobenthos [88].

According to [89], UV-Vis methods and fluorescence spectroscopy are commonly used due to their non-destructive working methodology and high sensitivity. The methodology of working with UV-Vis spectroscopy is presented, among others, by [90]. According to these authors, fluorescence and UV-Vis spectroscopy are well-suited for research on humic acids (or substances in general) due to their ability to emit light (including fluorescent) and allow for determining not only the concentration of these substances but also the molecular structure (and thus their identification) and structural transformations. Another interesting application of UV-Vis and fluorescence spectroscopy is presented by an analysis of humic compounds from soil [91]. It can therefore be concluded that using the methodology from [90,91], one can try to study the impact of surface runoff, or rather their humic fraction and their derivatives, on the ecosystem of a lake or, to a lesser extent, a river. The possibility of such use is mentioned in [90], where it is said that once knowing the composition of humic acids in the bottom sediment of a lake, it is possible to know the origin of organic matter on the bottom in a given place.

Nowadays, soil and water pollution with heavy metals is becoming a serious problem. Spectrophotometric methods can be used to investigate the origin of heavy metals (lead in this case) in soils. Such possibilities are indicated by [92], who claim that the isotopic composition of lead, along with its concentration, can answer the question of its origin in the environment.

Is it possible to directly or indirectly investigate the origin of heavy metals in waters using the methods presented in [92]? It probably is by comparing the content of heavy metals in soil, surface runoff, and sediments at the bottom of water bodies. One can also use humic substances, which, according to [90], have an affinity for metals.

UV-Vis is also related to the PARAFAC spectral analysis, which shows the peaks of individual wavelengths in three dimensions. This analysis can be performed in many programs, such as R [93] with the staRdom package [94] or in Matlab [95].

## 6. Semiconductors

In the field of modern electronics, FT-IR spectroscopy is used primarily for the qualitative assessment of the composition of semiconductor materials [96,97,98,99,100,101,102].

FT-IR can be a supporting tool in relation to using synchrotron radiation and NIR spectroscopy, which was shown in the example of studies on SiO_2_, SiC, Si_x_N_y_, and TiO_2_ [96]. Taking into account the complementarity of FT-IR and Raman spectroscopy, research based on the analysis of vibrational spectra is often enriched with information on electronic structures (UV-Vis) and nuclear dynamics (NMR). In addition, the results of empirical research are a reference point for theoretical calculations (Discrete Fourier Transform, DFT), as evidenced by the work documenting the SnO_2_ research [97].

Research on the doping of semiconductors with organic compounds should be considered interesting. Studies in this field for aniline, *m*-aminobenzoic acid and *p*-benzoquinone are summarised in the article [98]. The degree of doping of metallic spatial lattices was assessed using FT-IR.

The multifaceted work [99] on the use of CdS in the context of advanced protein studies, combining the use of mass spectrometry in the analysis of laser desorption products, is based on the use of FT-IR to assess the purity of CdS connections with 4-aminothiophenol and 11-mercaptoundecanoic acid.

In their study on the effectiveness of the use of TiO_2_ in the processes of creating thin organic coatings [100,101], FT-IR spectroscopy was used to assess the degree of their durability.

Finally, the semiconductor material itself can be assessed using FT-IR, as shown in the example of CuO in [102].

## 7. Food Industry

FT-IR technology has led to a renewed interest in using IR spectroscopy as a quantitative analytical tool, and with the technology’s price decreasing, it is now possible to consider using FT-IR for multi-component proximate analysis of foods. Fats, proteins, carbohydrates, and moisture can be simultaneously and conveniently measured by FT-IR in conjunction with ATR cell technology and could provide significant savings of cost and time for the food industry [103].

With the increasing globalisation of the food supply chain, several stakeholders in between “farm-to-fork” and the intrinsic vulnerability of such a system, food quality, and safety have become of increased concern for consumers, food producers, and governments (especially American Food & Drug Administration and European Commission). Scandals such as the addition of melamine in baby formula in China in 2008 [104], the European horse-meat scandal in 2013 [105], and the issue of peanuts and almonds found in ground cumin and paprika in Europe and the USA in 2015 [106], all illustrate the negative impacts of food fraud, including both adulteration and authenticity.

### 7.1. Food Adulteration

It can be more precisely defined as the process in which the quality of food is degraded on purpose—either by the addition of low-grade quality material or by extraction of valuable ingredients [107].

Food adulterations are performed to enhance the quantity of food, with a view of increasing the profit by adding some chemical substances which are not supposed to be the legal ingredient and are subsequently difficult to detect [108].

Nowadays, the most common food categories susceptible to any food fraud are in descending order: olive oil, fish, organic foods, milk, grains, honey and maple syrup, coffee and tea, spices, wine, and certain fruit juices [109].

The so-called premium food products, such as meat [110], spices and herbs [111], honey [112], olive oil [113], and coffee [114], are particularly susceptible to adulteration, especially when they are produced and supplied through complex supply chains [115].

Food adulteration is not only of global economic concern but can also have severe adverse health effects, such as the development of cramps, nausea, diarrhoea, vomiting, nerve damage, allergic reactions, and paralysis [116]. For instance, in 2015, there were reports of the addition of peanuts and almonds to cumin and paprika powder in the USA and Europe [106]. The unintentional ingestion of these two allergens could potentially lead to severe or even lethal allergic reactions [116]. It is, therefore, very important to investigate food fraud not only from an economic point of view but also from a health perspective.

The scandals concerning fraudulent foods have increased the pressure on food laboratories to develop fast and reliable screening methods for the detection of food fraud. One of the most commonly used screening techniques utilised by both industry and governmental laboratories for food fraud is FT-IR based on mid-infrared (MIR) vibrational spectroscopy [109], as it offers a rapid and reliable detection method. Over the last decades, FT-IR has proven to be a powerful tool for screening foods for adulteration and authenticity.

FT-IR analysis, in combination with multivariate data analysis, is sufficient to detect the level of adulteration. Food fraud is by no means a new phenomenon, but recent scandals have illustrated the vulnerability of the modern and increasingly global food supply chain. Developing rapid and easy screening techniques using FT-IR, for instance, is therefore imperative to ensure future food safety [117].

#### 7.1.1. Adulteration of Fats

Expensive high-quality vegetable oils and animal fats are susceptible to adulteration. The substitution of high-quality and expensive oils and fats with cheaper and inferior-quality oils or fats and labelling those products as pure are often used by producers to obtain a maximum profit [118].

Experimental results showed that even if the spectral differences in the middle infrared spectral region are very small because most vegetable oils contain the same type of fatty acids (especially those with C16 and C18) and triacylglycerols content is similar (C50, C52, and C54), there are subtle spectral differences in the spectrum of various types of vegetable oils. This allows for the detection of foreign oil addition in an oil sample using calibration curves established for certain characteristic frequencies in known mixed oils [118]. The example of data showing this phenomenon is presented in Table 3, and the spectra are graphically shown in Figure 7 and Figure 8.

Infrared spectroscopy, combined with chemometric, has been used for the detection of olive oil adulterated with hazelnut oil [119], sunflower and corn oils [120], sunflower oil [121], and corn, hazelnut, soybean, and sunflower oils [122].

FT-IR spectroscopy combined with multivariate calibrations and discriminant analysis can be used to monitor the adulteration of extra virgin olive oil with much cheaper oil, e.g., palm oil [113].

#### 7.1.2. Adulteration of Coffee

Coffee is considered one of the most common drinks all around the world. By combining ATR with FT-IR, it is possible to detect adulteration and multiple adulterants in roasted and ground coffee. Furthermore, the utilisation of data fusion (DF) to combine the data obtained by ATR-FT-IR and diffuse reflectance Fourier transform infrared spectroscopy (DRIFTS) resulted in an improvement of adulterant discrimination models. DF models were capable of detecting adulterated samples of coffee with coffee husks, spent coffee, roasted barley, and roasted corn and were also capable of identifying the adulterants, even if a mixture of up to four different adulterants was used [114].

#### 7.1.3. Adulteration of Spices

Adulteration of spices is an alarming trend, and the common adulterants are sand, dirt, artificial colour and starch, animal waste, chalk powder, sawdust, lead chromate, argemone seeds, and different type of chemicals. An incident was reported by the Times of India (Indian edition, 20 August 2017) when the health department had seized adulterated chilli, turmeric, and coriander stalks from their cold storage, the manufacturing unit using the inedible red colour to give redness to the chilli powder by using lower-quality chilli and wheat husk [123].

Not only does FT-IR provide the qualitative assessment of food, but also the detection of compounds available in food. In comparison with other detection techniques, FT-IR is rapid, reliable, and requires no sample preparation. There is a huge scope for the FT-IR to assess the quality of spices and herbs [123].

#### 7.1.4. Adulteration of Fruit Juices

According to the European Committee on the Environment, Public Health, and Food Safety, fruit juices are included in the top ten food products that are most at risk of food fraud [124].

FT-IR spectrometry combined with chemometric analysis of spectral data was able to uncover the adulteration of mango juice by sugar addition [125].

The authentication of Concord grape juice, among other grape varieties (red, white, and Niagara), in grape juice blends using FT-IR was studied by [126].

The potential adulteration of pomegranate juice concentrate with grape juice concentrate can be detected using FT-IR spectroscopy coupled with PCA [127].

The applicability of FT-IR in the detection of water and sugar addition to authentic orange juice was deeply discussed in [128].

#### 7.1.5. Adulteration of Wheat

Wheat (*Triticum aestivum* L.) is one of the most important cereal crops and is consumed as a staple food around the world. Vibrational spectroscopies, such as FT-IR, show great potential in testing wheat authenticity due to the instrumental evaluation and development of software advancements [129].

#### 7.1.6. Adulteration of Sugar

Sugar adulterants, such as plastics, chalk, urea, washing soda, sugar substitutes, and other sugar products, are used to increase the packaging weight, improve colour and taste, aid moisture absorption, increase profit, and thus make the cost of production lower [130].

The advantage of IR, which makes it better suited for adulteration analysis, is its non-destructive nature. FT-IR was used to detect adulteration in brown sugar that was intentionally doped with various concentrations of coconut sugar [131].

#### 7.1.7. Adulteration of Cow–Buffalo Milk

Soya milk (SM) is often used as an alternative to dairy milk due to its lower price and quite similar protein as cow milk, except for sulphur-containing amino acids, in which SM is deficient [132]. Soybean, together with milk, is one of the major allergenic foods, which mostly affects the infant population when the gut barrier is immature, and the immune system is still refining its ability to tolerate food proteins [133].

ATR-FT-IR, in conjunction with chemometric analysis of spectral data, is able to uncover the presence of SM in cow milk. There was a clear difference in the absorption values of SM and milk in the wavenumber region of 1639–1613 cm^−1^ [134].

### 7.2. Quality Assessment of Meat Products Using FTIR

Meat and meat products are very valuable but highly perishable. There is a need for reliable assessment techniques to ensure the safety and quality of these products throughout their shelf life. FT-IR spectroscopy coupled with chemometrics has drawn attention to quality control, safety assessment, and authentication purposes in the meat and meat products domain [135].

Meat is an essential food product, and an increase of 1.2% in global meat consumption by 2028 is expected. To meet increasing consumer expectations towards manufacturing safe, high-quality, sustainable, and cost-effective products, scientists and the meat industry are seeking alternative ways to ensure certain meat quality attributes with extended shelf life and storage stability [136].

Among several qualitative and quantitative methods, FT-IR spectroscopy has been employed for various quality and process control purposes in the food industry because it offers excellent opportunities for structural and functional studies. In addition to being fast, sensitive, and safe, FT-IR spectrometers provide simple and non-destructive measurements without the need for complicated, time-consuming sample preparation. Figure 9 summarises the applications of FT-IR spectroscopy for the analyses of meat and meat products.

### 7.3. Microbial Quality Assessment of Minimally Processed Fruits Using FT-IR

Market demand for minimally processed fruits has seen a huge boom in recent years due to convenience and changes in consumer attitude [137].

Microbial activities that lead to spoilage cause the development of off-odour, off-flavour, and slime formation, which makes a product unfit for human consumption. Microbial quality is the critical parameter determining the safety of refrigerated perishables [138].

#### 7.3.1. Minimally Processed Pineapple

The results of microbial quality assessment of minimally processed pineapple using GCMS and FT-IR indicate that models built using FT-IR data provided good prediction with low SEP and high accuracy. These results suggest the possibility of using FT-IR for rapid prediction of microbial quality [138].

#### 7.3.2. Minimally Processed Pomegranate

Results indicate that FT-IR, along with supervised learning algorithms such as partial least square regression (PLS-R) and artificial neural networks (ANN), can be employed for the rapid determination of microbial quality in minimally processed pomegranate. The advantage of this technique is its rapidity and the convenience it adds when compared with conventional methods used for determining microbial quality. The expensive and time-consuming microbial analysis could be replaced by an online system based on spectroscopic data. FT-IR spectroscopy serves as a metabolic snapshot of stored, packed, and minimally processed pomegranates [137].

### 7.4. Microplastics

Microplastics (MPs) are defined as small particles of plastic material, commonly considered smaller than 5 mm [139], that could have been either purposely manufactured with that size (primary MPs) or resulted from the fragmentation of all types of plastics (secondary MPs). Continuous dumping of plastic causes a constant increase in MPs in oceans and seas, where aquatic organisms accumulate MPs in their guts [140]. As the size of MPs particles is very small, they can be ingested by a wide range of organisms that mistake them as food for prey [141], and through the aquatic animals, MPs can end up on our tables [140].

FT-IR spectroscopy has been extensively used in MP pollution research since 2004 [142]. It can identify all the molecular and functional groups present in plastic polymers [143].

FT-IR spectroscopy deals with the measurement of infrared (IR) radiation absorbed by an MP sample, allowing the study of molecular composition. An infrared spectrum represents a fingerprint of an MP sample with absorption peaks corresponding to the frequencies of vibration of atoms and molecules making up the material. Because each different polymer material is a unique combination of atoms, no two compounds produce the same infrared spectrum. Therefore, the chemical structure of a polymer molecule can be determined by FT-IR [144].

ATR technique coupled with FT-IR spectroscopy is widely used to characterise the large size MPs, whereas smaller MPs require the use of mFT-IR coupled with a detector. In particular, mFT-IR coupled with a focal plane array detector facilitates a much faster generation of MP chemical imaging by simultaneously scanning several thousand spectra within a single measurement. Moreover, the FT-IR method is used to understand the ecological effects of ingested MPs and their associated pollutants and biochemical variations at the cellular level [142].

## 8. Criminalistics

In the field of forensics, FT-IR spectroscopy provides valuable analytical services [145,146,147,148,149,150]. Examples include analyses of soils [145], cosmetics [146], paints and varnishes [147], drugs [148], inks [149], and hygiene products [150]. Routine data processing tools based on discriminatory models are used in all these works.

## 9. Museology

The use of FT-IR spectroscopy in the field of knowledge about art can be roughly divided into the assessment of the authenticity of paintings, relatively focused on assessing their age, analyses in the field of examining the composition (age) of excavations (ceramic materials and others), as well as covering the issues of nature conservation [151,152,153,154,155].

## 10. Conclusions

The wide range of applications of FT-IR that are presented in a synthetic approach unquestionably shows the importance of this technique in the current scientific and analytical research. Presentations of achievements with the use of other spectroscopic tools (UV-Vis and others) against FT-IR allow not only the highlights of its advantages but, above all, the granting of the preliminary assessment and the prospective of using this tool to researchers.

To sum up, FT-IR is a method with many advantages (easy preparation for analysis, effectiveness in detecting changes at the level of cell biochemistry, and precision in determinations). It can be efficiently used to identify chemical compounds by recognising their structure.

Further development of gas analysers using FT-IR spectroscopy may, in the future, significantly broaden the scope of applications for this equipment. In the case of greenhouse gas flux measurements, the potential increase in the number of gas concentration measurements per time (speed increase) would allow for applying this equipment in other analytical fields (e.g., in the micrometeorological eddy covariance method).

Finally, in light of the cited results of experimental research on model molecular systems forming molecular crystals stabilised by hydrogen bonds (dicarboxylic acids and others), the necessity of developing this branch of spectroscopy, also in the context of instrumental evolution, was confirmed. It must not be forgotten that the rapid development of IR spectroscopy, lasting over half a century, is dictated by the focus of researchers’ attention on one phenomenon—hydrogen bonds.

## Figures and Tables

**Figure 1 ijms-24-01013-f001:**
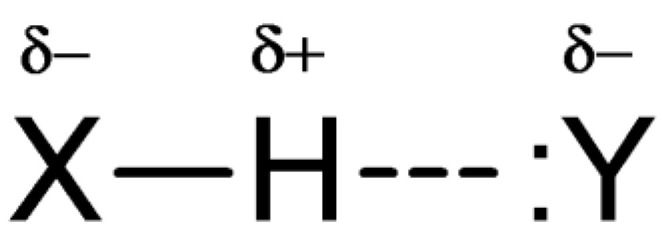
Isolated hydrogen bond.

**Figure 2 ijms-24-01013-f002:**
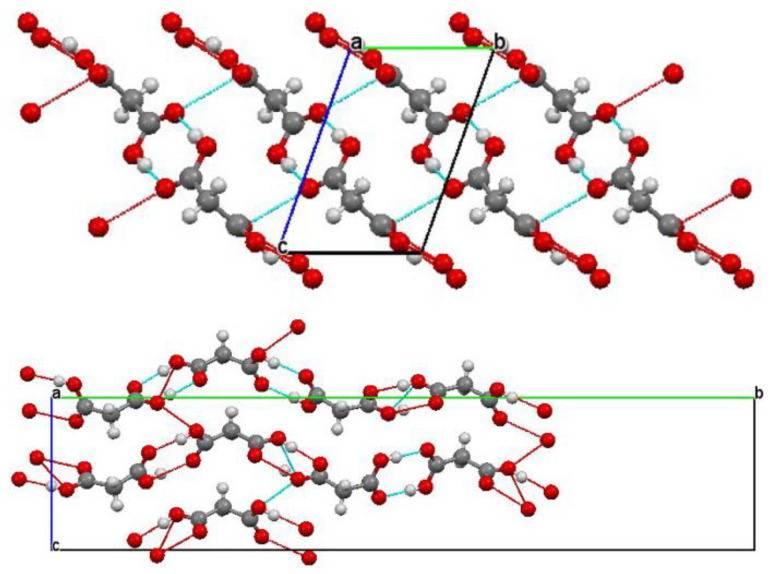
Lattice projection of malonic acid crystals [35].

**Figure 3 ijms-24-01013-f003:**
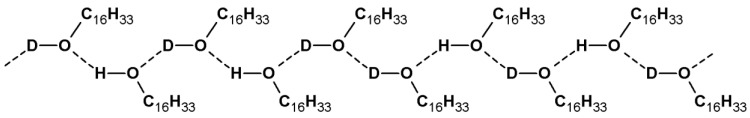
Cetyl alcohol isotopically diluted with deuterons.

**Figure 4 ijms-24-01013-f004:**
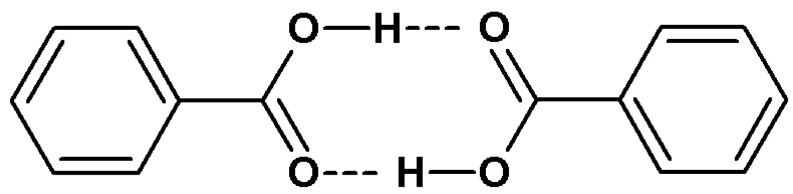
Dimerised form of isotopically neat benzoic acid.

**Figure 5 ijms-24-01013-f005:**
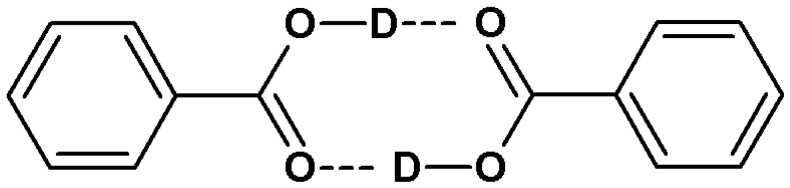
Deuterium-bonded dimers of benzoic acid.

**Figure 6 ijms-24-01013-f006:**
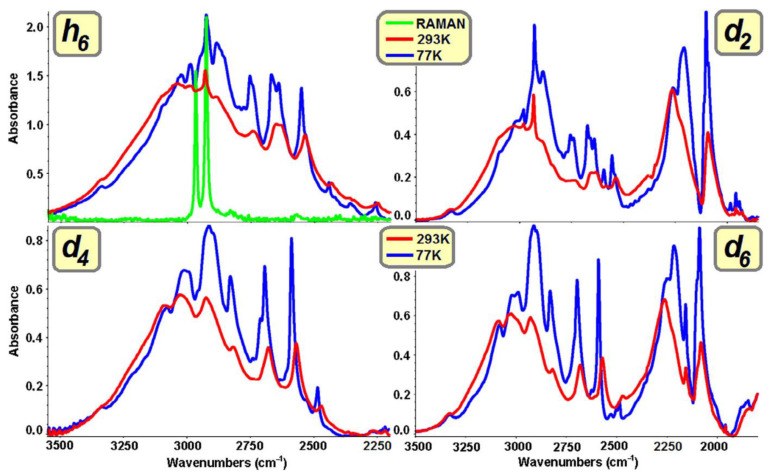
Infrared and Raman spectra of polycrystalline samples of succinic acid isotopic varieties, with a visible distinction between the ν_X–H_ and ν_X–D_ bands [27].

**Figure 7 ijms-24-01013-f007:**
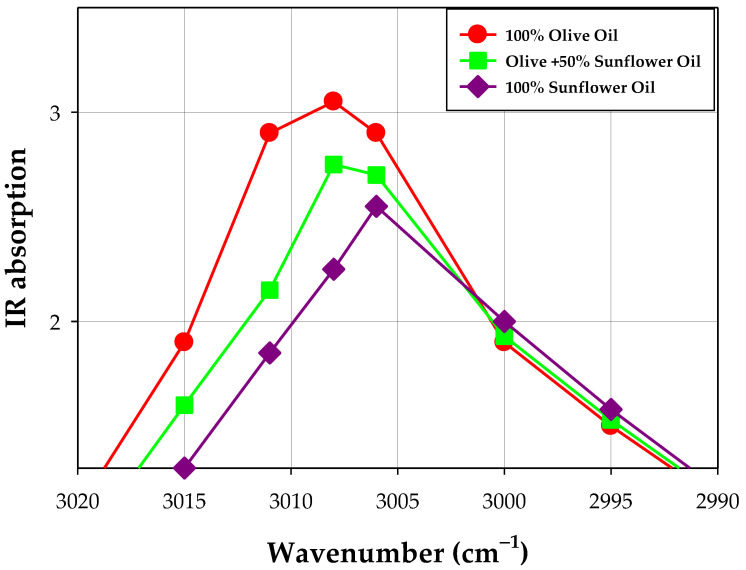
IR absorption spectra for olive and sunflower oils and their 50–50 mixture.

**Figure 8 ijms-24-01013-f008:**
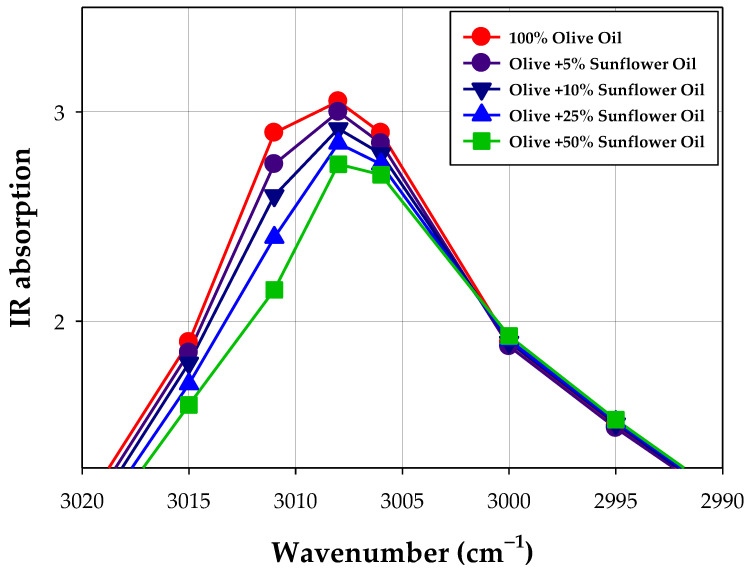
IR absorption spectra for pure and adulterated olive oil.

**Figure 9 ijms-24-01013-f009:**
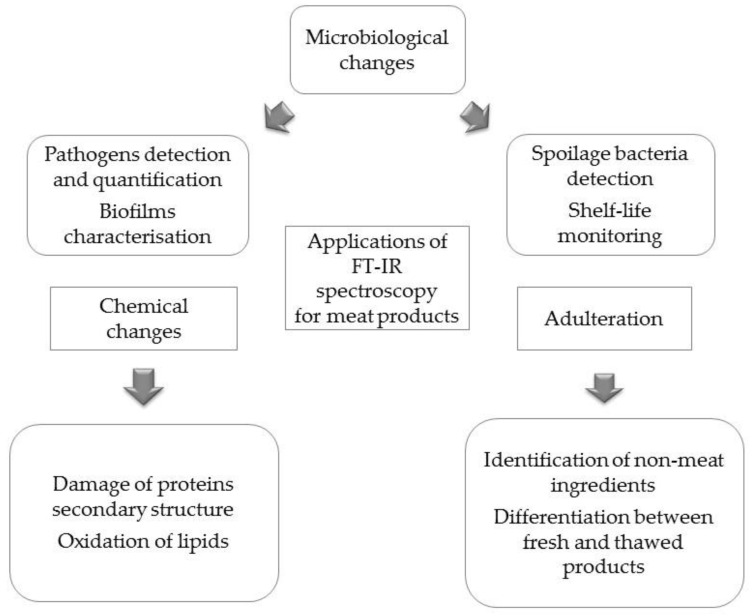
FT-IR spectroscopy applications for meat products.

**Table 1 ijms-24-01013-t001:** Energy characteristics of normal vibrations of hydrogen bonds.

Vibration	ν_X–H_	ν_X···Y_	δ_X–H_	δ_X···Y_	γ_X–H···Y_
*E* [cm^−1^]	1750 < *E* < 3700	50 < *E* < 650	1700 < *E* < 1800	*E* < 50	450 < *E* < 950

**Table 2 ijms-24-01013-t002:** Wavenumber ranges for classes of chemical compounds (according to [83], modified).

Chemical Class/Element	Wavenumbers [cm^−1^]
Carbohydrates	880–1064
Phosphorus	1190–1350
Amida II	1480–1575
Amida I	1575–1705
Lipids	1708–1780

**Table 3 ijms-24-01013-t003:** IR absorption for olive and sunflower oils and for their mixture according to [118], modified.

Sample	Wavenumber [cm^−1^]								
	1753	2990	2995	3000	3006	3008	3011	3015	3020
Olive oil	3.83	1.15	1.50	1.90	2.71	2.38	1.64	1.90	1.10
Olive + 5% sunflower oil	3.16	1.14	1.49	1.88	2.29	2.44	1.67	1.85	1.05
Olive + 10% sunflower oil	3.56	1.15	1.51	1.90	2.45	2.56	1.71	1.80	1.00
Olive + 25% sunflower oil	3.58	1.16	1.52	1.91	3.13	2.74	2.21	1.70	0.95
Olive + 50% sunflower oil	3.54	1.17	1.53	1.93	3.33	2.96	2.61	1.60	0.90
Sunflower oil	3.47	1.20	1.58	2.00	2.20	2.35	2.75	1.30	0.70

## Data Availability

Not applicable.

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
