# Peer review of "The Analytical Possibilities of FT-IR Spectroscopy Powered by Vibrating Molecules"

_ijms, 2023, doi:10.3390/ijms24021013_

Round 1

Reviewer 1 Report

Dear Authors, 

The manuscript ijms-2130516, entitled 'The Analytical Possibilities of FT-IR Spectroscopy Powered by Vibrating Molecules' exposes the state of progress in the development of spectroscopic methods based on the use of mid-infrared radiation in the context of applications in various fields of science and technology. The authors drew attention to the most important solutions specific to both spectroscopy itself (ATR technique) and chemometric data processing tools (PCA and PLS models). 

The manuscript is well organized, with the presentation of the methods involving FTIR, as well as the direct applications where this technique is successfully used. The exposed methods are well argued and widely debated. The tabulated data are consistent and the figures are of good quality. However, figure 7, page 24, should have the scales on both coordinate axes written in a font close to that of the manuscript text, with better differentiation of the curves (with different line type) for better visualization. The conclusions are concise and supported by the results and discussions in the manuscript.

I propose the acceptance for publication of the manuscript, with the small corrections on figure 7, in its current form.

I accept in the present form.

Author Response

Yours faithfully –

Eliza Gruczyńska-Sękowska

Reviewer 2 Report

In general, the article is very well written and it is recommended to be published in the journal, Please, the authors fix the following general flaws (Minor Revise):

In the abstract of the article, you should include the innovation of your work and the purpose of the research currency. Use more keywords. The number of references is very high. Reduce drastically. Also, use this recommended reference: doi.org/10.1016/j.ejmech.2022.114443

Use more graphs in the results section. The English language of the article has problems in some sentences, Fix them.

Author Response

(The authors gave the same response as above.)
